# Early psychological interventions for prevention and treatment of post-traumatic stress disorder (PTSD) and post-traumatic stress symptoms in post-partum women: A systematic review and meta-analysis

P. G. Taylor Miller[1], M. Sinclair[1]\*, P. Gillen[1,2], J. E. M. McCullough[1], P. W. Miller[1,3], D. P. Farrell[4], P. F. Slater[1], E. Shapiro[5], P. Klaus[6,7]

1 Institute of Nursing and Health Research, Faculty of Life and Health Sciences, Ulster University, Belfast, Northern Ireland, United Kingdom, 2 Southern Health and Social Care Trust, Craigavon, Northern Ireland, United Kingdom, 3 Mirabilis Health Academy, Newtownabbey, Northern Ireland, United Kingdom, 4 Department of Violence Prevention, Trauma and Criminology, School of Psychology, University of Worcester, Worcester, England, United Kingdom, 5 EMDR Association, Mental Health Centre, Lev HaSharon, Israel, 6 Dona International, Chicago, Illinois, United States of America, 7 PATTCh, Prevention and Treatment of Traumatic Childbirth, Seattle, Washington, United States of America

\* m.sinclair1@ulster.ac.uk

## Abstract

### Background

Pre-term or full-term childbirth can be experienced as physically or psychologically traumatic. Cumulative and trans-generational effects of traumatic stress on both psychological and physical health indicate the ethical requirement to investigate appropriate preventative treatment for stress symptoms in women following a routine traumatic experience such as childbirth.

### Objective

The objective of this review was to investigate the effectiveness of early psychological interventions in reducing or preventing post-traumatic stress symptoms and post-traumatic stress disorder in post-partum women within twelve weeks of a traumatic birth.

### Methods

Randomised controlled trials and pilot studies of psychological interventions preventing or reducing post-traumatic stress symptoms or PTSD, that included women who had experienced a traumatic birth, were identified in a search of Cochrane Central Register of Randomised Controlled Trials, MEDLINE, Embase, Psychinfo, PILOTS, CINAHL and Proquest Dissertations databases. One author performed database searches, verified results with a subject librarian, extracted study details and data. Five authors appraised extracted data and agreed upon risk of bias. Analysis was completed with Rev Man 5 software and quality of findings were rated according to Grading of Recommendation, Assessment, Development, and Evaluation.

**Data Availability Statement:** All relevant data are within the paper and its Supporting Information files.

**Funding:** No funding was received to complete this work. PGTM received a scholarship from the Department of the Economy, Northern Ireland to complete PhD study.

**Competing interests:** I have read the journal's policy and the authors of this manuscript have the following competing interests: PWM is clinical lead for Mirabilis health a private trauma-focused mental health service, which provides pro bono psychotherapy to participants for research projects. PWM provides EMDR training as a part of Mirabilis Health Academy. PWM is a member by invitation of the Council of Scholars; part of the Future of EMDR Therapy Project. The Project works on developing global EMDR standards for training and competency benchmarking guidelines in EMDR therapy. ES is the developer of the EMDR G-TEP protocol and provides training workshops for EMDR clinicians and is also a member of the Council of Scholars.

## Results

Eleven studies were identified that evaluated the effectiveness of a range of early psychological interventions. There was firm evidence to suggest that midwifery or clinician led early psychological interventions administered within 72 hours following traumatic childbirth are more effective than usual care in reducing traumatic stress symptoms in women at 4–6 weeks. Further studies of high methodological quality that include longer follow up of 6–12 months are required in order to substantiate the evidence of the effectiveness of specific face to face and online early psychological intervention modalities in preventing the effects of stress symptoms and PTSD in women following a traumatic birth before introduction to routine care and practice.

## Prospero registration

CRD42020202576, https://www.crd.york.ac.uk/prospero/display_record.php?RecordID=202576

## 1.0 Introduction

Prevalence rates for Post-Traumatic Stress Disorder (PTSD) range from 3.1 to 15.7% of postpartum women [1]. It is reported that 1 in 10 women experience PTSD at 4–6 weeks postpartum [2]. Other women report symptoms of PTSD, including re-experiencing, avoidance, emotional numbing, hyper arousal and negative changes in thinking and mood in the immediate period following childbirth, but do not qualify for the disorder itself. Recent research has found that women are particularly vulnerable to re-experiencing symptoms of PTSD during the perinatal period [3]. Symptoms such as these impact upon women's quality of life, physical health, personal relationships and the infants physical, behavioural, social and emotional development [4–7]. PTSD following birth is highly co-morbid with depression [8], and there has been an emergence of investigation in latent profiling of mental health disorders and co-morbidity in specific populations experiencing PTSD [9]. Quantitative and qualitative differences in symptom profiles highlight the importance in the design, development and implementation of interventions for presenting populations and their subgroups.

### 1.1 Description of PTSD

The nosology of PTSD has undergone a change in recent years with further development of diagnostic models endorsed by the Diagnostic Statistical Manual for psychological disorders [10] and the International Classification of Disease [11]. PTSD, previously set within the criteria of anxiety disorder, now rests within the category of stress disorders, alongside other disorders that are characterised as being triggered by the onset traumatic event.

Nice (2020) define traumatic birth as a criterion 'A' qualifying event for PTSD in women;

"Traumatic birth includes births, whether preterm or full term, which are physically traumatic (for example, instrumental or assisted deliveries or emergency caesarean sections, severe perineal tears, postpartum haemorrhage) and births that are experienced as traumatic, even when the delivery is obstetrically straightforward" [12].

This description fits with ICD-11 core diagnostic features of PTSD as "A disorder that develops following exposure to an extremely threatening or horrific event or series of events" and DSM-5 criterion A, that states "Exposure to actual or threatened death, serious injury or

sexual violation from, directly experiencing the traumatic event, witnessing the traumatic event in person, learning that the traumatic event occurred to a close family member or close friend (with the actual or threatened death being either violent or accidental); or the individual experiences first-hand repeated or extreme exposure to aversive details of the traumatic event (not through media, pictures, television or movies unless work-related)" [13].

Secondary tokophobia (fear of childbirth) has been equated to PTSD following childbirth, as it is caused by a previous traumatic birth. Women who experience secondary tokophobia are more likely to elect for birth by caesarean section in subsequent pregnancies. A previous longitudinal study found that spontaneous recovery from post-traumatic stress following childbirth is unusual [14, 15].

## 1.2 Description of the intervention

There are a broad range of interventions targeting post-traumatic stress disorder following childbirth including psychotherapeutic therapy, grief counselling, expressive writing, midwifery counselling and debriefing. Traditionally, midwifery led psychological debriefing is offered to women in the days and weeks following the birth of their baby. It has been described as an opportunity for the mother to describe her experience, express her emotions and feelings in relation to the negative event and "fill in the gaps" [16]. Evidence of the effectiveness of trauma focused debriefing is controversial with systematic literature reviews concluding that there are no midwifery interventions aimed at reducing PTSD that can be recommended to clinical practice following childbirth [16–20]. NICE [12] and the International Society for Stress Studies [21] conclude that there is insufficient evidence to recommend psychological debriefing for the prevention of PTSD. Providing an ineffective treatment may be regarded as harmful, as reliving traumatic experiences can retraumatise an individual and deter them from appropriate treatment interventions.

Early intervention treatments are administered in the days, weeks and first three months of a traumatic event. This early period following a traumatic event is critically important in the development of clinically significant PTSD symptoms as the traumatic memory remains fragmented [22–24]. Early psychological intervention may prevent the accumulation of unprocessed traumatic memories, as re-current exposure to trauma may further sensitise underlying and dormant disorders [25].

NICE [12] recommend Trauma Focused CBT and prolonged exposure therapies up to four weeks following a traumatic incident, and Eye Movement Desensitisation and Reprocessing (EMDR) if the patient requests it after four weeks of the onset event. The International Society for Traumatic Stress Studies [21] advise on multiple session trauma focused cognitive behavioural therapy and EMDR as the standard recommended early treatment interventions for PTSD symptoms in adults. Table 1 describes prominent early interventions for PTSD in adults.

A recent Cochrane review suggests that there is insufficient evidence to recommend the introduction of any universal multiple session early intervention (offered to all individuals exposed to a traumatic event regardless of symptomology) to prevent PTSD to clinical practice [30]. An earlier Cochrane review assessed the effectiveness of early psychological interventions in treating sub-threshold or full diagnostic status for Acute Stress Disorder (ASD) or acute PTSD in adults. Results concluded that there was evidence of the effectiveness of Trauma Focused Cognitive Behavioural Therapy (TF-CBT) in reducing acute traumatic stress symptoms when compared to waiting list and supportive counselling interventions [31]. Another recent systematic review and meta-analysis [32] report the benefits of early EMDR intervention, trauma-focused cognitive-behavioural therapy (TF-CBT) and cognitive therapy without exposure in adults who were screened for post-traumatic stress symptoms following a

**Table 1. Description of early treatment interventions for post-traumatic stress disorder.**

| Intervention | Theoretical Framework | Length of treatment | Intervention Description | Components |
|---|---|---|---|---|
| **Trauma Focused Cognitive Behavioural Therapy (TF-CBT) Deblinger E, Cohen J, Mannarino A (2016)** [26] | Cognitive Behavioural & Social Learning Theory | 8–25 sessions. | Cognitive behavioural techniques are structured and utilised to help modify distorted or intrusive thinking and negative behaviours in response to a traumatic event. | 8 components: Parenting skills/psychoeducation, relaxation techniques, affective expression and regulation, cognitive coping, development and processing of a trauma narrative, gradual exposure, conjoint parent/child sessions, safety review and future development. |
| **Exposure Therapy (Foa & Rothbaum, 1998)** [27] | Cognitive behavioural theory | 8–15 sessions | Individuals are "exposed" to the objects, activities, situations they fear and avoid in a safe environment; reducing fear and avoidance. | Paced in various ways: **Graded exposure**: based on an agreed fear hierarchy in which fears are ranked according to difficulty. **Flooding**: Using the exposure fear hierarchy to begin exposure with the most difficult tasks. **Systematic desensitization** Exposure is combined with relaxation exercises which may enable the person to associate the fear with relaxation. |
| **Eye Movement Desensitisation and Reprocessing (EMDR) Shapiro, F (1989)** [28] | Adaptive Information Processing Theory | 1–30 sessions. | Integrative transdiagnostic therapy Addresses the past, present, and future aspects of a dysfunctional memory. Dual attention process of recalling distressing events while receiving 'bilateral sensory input' including eye movements, hand tapping, auditory tones. | 8 phase approach: History taking, client preparation, assessment, desensitization, installation, body scan, closure re-evaluation of treatment effect. |
| **Midwifery Counselling** | Cognitive behavioural theory in some models | 1–6 sessions | The midwife encourages the mother to describe and talk about her unpleasant experiences, along with feelings and emotions in relation to the negative event. | Varies Emphasis is on a supportive counselling relationship. Cognitive restructuring techniques help mothers recognize and change their dysfunctional attitudes towards childbirth and its related pain (Gamble) [29] |
| **Midwifery postnatal Debriefing** | No theoretical framework | 1 session | Discussion of the birth experience, provision of further information and providing answers to clinical questions relating to the birth and breastfeeding the infant. | The midwife supports and listens to the woman's concerns, feelings, expression of birth experience, answer questions relating to the birth and provides information requested by the women. |

traumatic experience. The greatest effects were found in adults who were diagnosed with ASD and PTSD, demonstrating the importance of providing early intervention and treatment to those in clinical need.

A recent Meta-analysis [33] found moderate treatment effect in favour of trauma focused psychological therapy compared to care as usual in treating post-traumatic stress symptoms in women following childbirth (SMD = -0.50 95% CI -0.73, -0.28 studies = 6, No = 296:305) with moderate heterogeneity between studies; $I^2$ = 40%. Another literature review found insufficient evidence to estimate the effectiveness of universal interventions in the primary or secondary prevention of PTSD or post-traumatic stress symptoms following childbirth [34]. De Bruijn [35] concluded that there is insufficient evidence to affirm the effectiveness of interventions targeted at treatment of PTSD in women following childbirth.

This review focuses on analysis of early interventions, delivered within 12 weeks for women who have experienced a traumatic birth.

### 1.3 Objectives

This review aims to answer the research question:

What are the effects of early psychological interventions delivered during the perinatal period on post-traumatic stress disorder and post-traumatic stress symptoms in post-partum women following a traumatic birth?

This review will provide evidence of the effectiveness of early psychological interventions in the prevention of PTSD and treatment of PTSD symptoms in women who have had a traumatic birth.

## 2.0 Methods

This review was conducted with adherence to the systematic review protocol registered in Prospero in accordance with PRISMA-P checklist and in accordance with the PRISMA (2009) checklist which can be found in supplementary files. Randomised controlled trials and pilot studies were eligible for inclusion in this review. Systematic reviews, meta-analysis, case reports, studies focusing on the effectiveness of spiritual intervention and interventions that focused on parenting skills were excluded from the review as were solely dismantling studies. Studies were included irrespective of sample size, language and publication status.

Studies that included adult women over the age of 18 years who had a full term or preterm traumatic birth experience were eligible for inclusion. Pregnant women were included for primary prevention or postpartum for secondary prevention (i.e., up to 3 months). There were no restrictions on diagnosis or comorbidity at the time period when the early intervention was administered. Assessment by DSM-III, DSM-III-R, DSM-IV, DSM-5, ICD-9, ICD-10 or ICD-11 diagnostic criteria for PTSD and Criterion A qualifying as experience of a psychologically traumatic birth were included.

### 2.1 Experimental interventions

We considered any experimental non-pharmaceutical intervention designed to prevent, reduce or treat symptoms of post-traumatic stress disorder delivered by one or more healthcare professionals or layperson, during the perinatal period as a primary or secondary early intervention beginning no later than three months after the traumatic event.

### 2.2 Comparator interventions

1. A control intervention that included Care as Usual, waiting -list control, minimal or placebo condition.

2. Any alternative trauma focused or non-trauma focused early psychological or biopsychological intervention.

### 2.3 Primary outcomes

Severity of post-traumatic stress symptoms using a validated and standardised self-report or clinician administered psychometric measure or diagnostic interview as stipulated by the Diagnostic and Statistical Manual of Mental Disorders (DSM-III, 1980; DSM-III-R, 1987; DSM-IV; 1995; or DSM-V, 2013, APA 2018) and the International Classification of Diseases (ICD-9, ICD-10 or ICD-11, WHO 1993, 2022).

## 2.4 Search methods for identification of studies

This review adheres to PRISMA reporting guidelines [36] and search criteria suggestions by the Cochrane Handbook [37].

## 2.5 Electronic database search strategy

The databases were searched using a sensitive set of relevant free text terms specifically related to the PICOS framework (see Table 2) Boolean operators were used for each database search with truncation and wildcards. The following set of search terms were utilised for searching the OVID database (please see supplement for specific database search terms). Truncation was adjusted for each specific search library, database and journal and grey literature provider.

**2.5.1 Electronic databases searched.** Searches of electronic databases; Embase, PsychInfo, AMED (Allied and Complementary Medicine), MEDLINE, through the OVID database and the Global Health Library. Searches of CINAHL, Published International Literature on Traumatic Stress (PILOTS) and Dissertations and Thesis through Proquest, The Lancet and The Cochrane Library were also performed. The World Health Organisation International Clinical Trials Registry Platform and ClinicalTrials.gov were included in the search and hand searched bibliographies of the included studies and for citations of articles were performed. An expert subject librarian provided validity for the searches up to 1$^{st}$ May 2020 to ensure reliability and rigour in accordance with PRESS guidelines [38].

## 2.6 Data collection, analysis and synthesis

Searches were saved on databases and references were transferred to Refworks Legacy version from each database and provider. A folder was developed for references saved from each database and provider in Ref works. 34 exact and close duplicates were removed. The review author independently read the abstracts of all studies. Study protocols were retained if available and saved in a separate protocol folder for cross referencing during the final review analysis. If an abstract described a pilot or RCT, review author independently read the full report to assess whether the study met the inclusion criteria. Five other review authors were then consulted on this process through discussion, and confirmation of consensus ensuring reliability of results.

A random effects model was used for comparisons as it was anticipated that there would be a wide range of interventions included in the analysis. Studies were assessed qualitatively for heterogeneity in terms of the variability in the participants, interventions and outcomes. Subgroup analysis, sensitivity analysis and investigation of heterogeneity ($I^{2)}$ was conducted by type of intervention. Principle summary measures were risk ratio and difference in means.

**Table 2. PICOS framework.**

| | |
|---|---|
| Population | (perinatal OR postnatal OR antenatal OR prenatal OR pre-natal OR ante-natal OR perinatal OR birth OR childbirth OR parturition OR postpartum OR caesarean OR caesarean OR haemorrhage OR assisted delivery OR vacuum delivery OR perineal tear OR stillbirth OR stillborn OR forceps OR instrumental delivery).af. |
| Intervention & comparison | (EMDR OR (eye movement desensiti#ation and reprocessing) OR CBT OR cognitive behavio?ral therapy OR iCBT OR online intervention OR telehealth OR exposure OR counselling OR ounselling OR therapy OR psychoeducation OR early intervention OR group intervention OR psychological OR psychotherapy OR debriefing OR rewind OR birth afterthoughts OR TF?CBT OR CPT OR cognitive processing therapy OR stabili#ation OR treatment as usual OR care as usual OR cau).af. |
| Outcome | (Post?Traumatic Stress Disorder OR PTSD).af. |
| Study Design | (RCT or randomi#ed control* trial or protocol or pilot or clinical trial).af. |

All studies included outcomes of PTSD symptoms and or prevalence and or diagnosis in adult women who had experienced a physically or psychologically traumatic birth. Studies were clinically diverse in terms of the type of interventions assessed. Meta-analysis was conducted, and results considered in order to fully analyse the data which may have provided important insights. Outcomes were evaluated by Grading of Recommendation, Assessment, Development, and Evaluation' (GRADE) approach [39]. GRADE guidelines framework was used to assess the quality of evidence in terms of study limitations, inconsistency/ unexplained heterogeneity, indirectness of the available evidence, imprecision of effect estimates. High, moderate and low specifiers were applied to each comparison as an indication of the confidence that the effect estimate would remain unchanged as a result of further research.

## 3.0 Results

### 3.1 Flow of Studies

The study selection process can be viewed in the PRISMA flow diagram (Fig 1) of included studies in this review. A total of 785 references were identified from data bases and a further six studies from hand searching. 757 studies remained after de-duplication. Review authors independently screened the titles and abstracts of the records. 674 studies were excluded as not relevant as they did not meet the inclusion criteria. Four references were for ongoing studies.

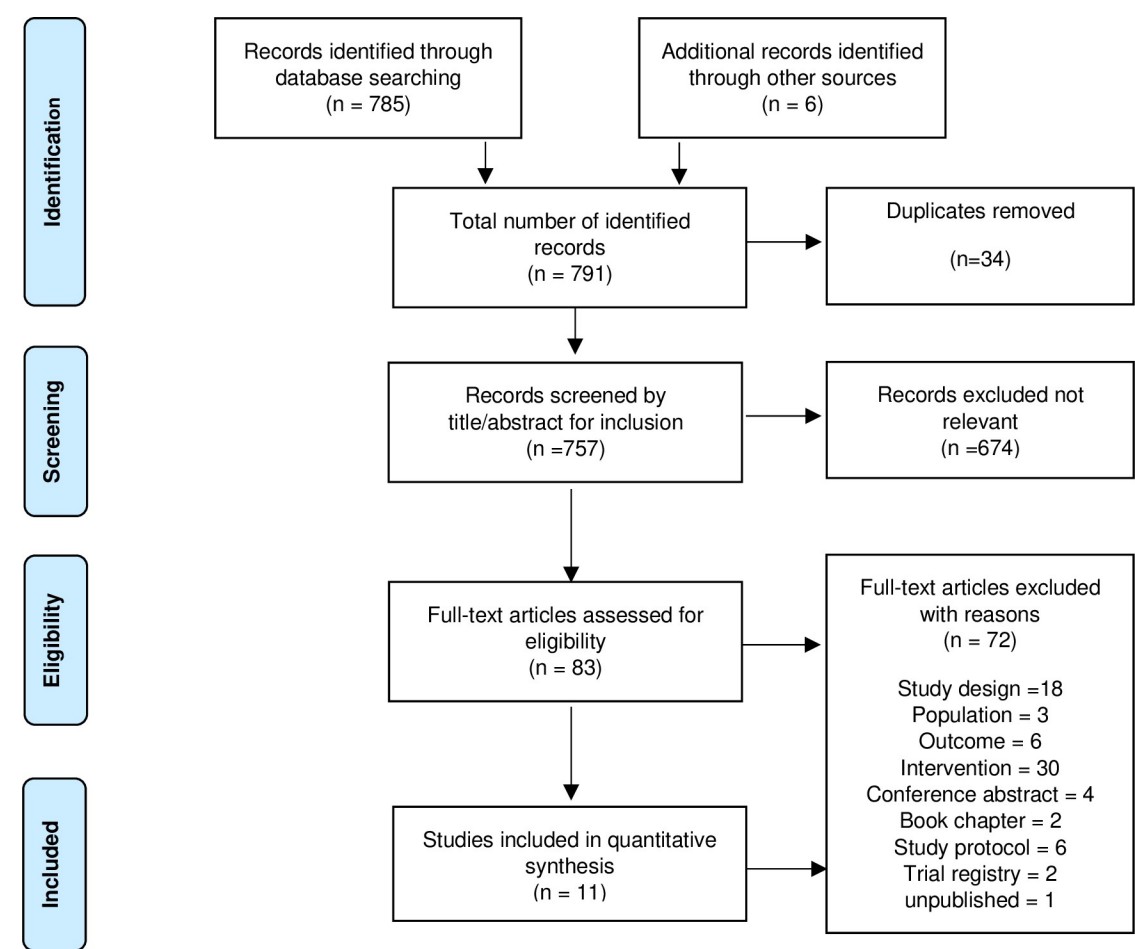

**Fig 1. PRISMA flow diagram of included studies.**

Full text papers were retrieved for the remaining 83 studies. 72 studies were excluded as they did not meet the eligibility criteria. A total of 11 studies were found to meet the specified eligibility criteria.

## 3.2 Included studies

There are a total of 11 studies in this review, with characteristics as follows. (Please also refer to characteristics of included studies Table 3 for study design, number of participants, setting, dosage and timing of the early intervention and psychometric PTSD measures used). One study could not be accessed and is awaiting classification [40].

**3.2.1 Participants.**   There was a total of 1,875 participants across eleven studies. Nine of the eleven studies delivered interventions to women within one week following a traumatic birth. The remaining two were delivered to women within one month of exposure to a traumatic birth irrespective of psychological symptomology. Six of the eleven studies assessed preventative psychological interventions offering psychological support to women who experienced a psychologically traumatic birth as specified by DSM-IV and DSM-V Criteria A [29, 41–44, 49]. Please refer to these afore mentioned papers for detailed screening questions for qualifying for a psychologically traumatic birth. Three studies screened women for clinically significant symptoms of stress; rated by the Parental Stress Scale NICU [48], scores over 24 on the IES-R [44], and depression; as measured by scores over 10 on the Edinburgh Postnatal Depression Scale [43]. Three studies assessed the effectiveness of psychological interventions in reducing PTSD in women who had experienced a physically traumatic birth, irrespective of mental health profiles [46, 45, 47]. Two studies recruited women whose babies were receiving care in the Neonatal Intensive Care Unit [48, 50].

**3.2.2 Interventions.**   Eleven studies compared the effects of early psychological interventions with care as usual on PTSD in women who had experienced a physical or psychologically traumatic birth. The studies included a variety of psychological interventions and are categorised in Fig 2.

Seven studies assessed immediate response early psychological interventions characterised as rapid access and low dosage; delivered by psychologists, paraprofessionals, nurses or midwives via face to face, online, group or self-help materials. One study tested the effectiveness of a group counselling intervention consisting of two sessions delivered one to two months following caesarean section by an obstetrician and midwife [46]. Three Studies assessed high intensity psychological interventions with a higher number of sessions delivered by psychologists or specialist mental health practitioners. These included six sessions of trauma focused cognitive behavioural therapy delivered by Doctoral students in training vs an active psychoeducation control [48]; seven sessions of the "Care and Cues" intervention aimed at providing parents with the skills to interpret and respond to their premature infant vs an attentional intervention without the behavioural education delivered by a psychologist or nurse [50] and four sessions of counselling and psychoeducation delivered by an obstetric doctor with a primary psychotherapy qualification vs usual care condition [47].

## 3.3 Summary assessments of Risk of Bias

Risk of Bias was assessed in accordance with the criteria outlined in the Cochrane Handbook for Systematic Reviews of interventions [37]. Bias was assessed as high, uncertain, or low risk through review team consultation and consensus met. Summaries of the risk of bias for each study are stated in the risk of bias Table in Fig 3. The risk of bias judgements were taken into account in the final consideration of treatment effect. Studies with high risk of bias were marked down by 1.

**Table 3. Study characteristics table.**

| Author | Country & Setting | Study Design No of Participants/ Interventions | Trauma Exposure | Intervention, Dosage & Timing | Comparison | Follow-up | Outcome Measures | Adverse Effects Participant Subjective Experience | Results | Comments |
|---|---|---|---|---|---|---|---|---|---|---|
| Abdollahpour (2019) [41] | Hospital Iran | RCT 3xarm Parallel design N = 193 Cont 86: CBT 53: De-Bf 54 PP Cntl = 81 CBT = 47 Debrief = 51 | (Inc) Experienced a traumatic birth within the previous 48hrs, Qualified for PTSD criterion A DSM-5 as assessed by screening scale. | Face to Face Debriefing, cognitive behavioural counselling Dosage: 1 x 40-60min administered within first 48hrs following birth. Mother provided with counsellor's phone number. Interventions delivered by midwife (MSc. midwifery counselling). | 3 x groups Cognitive behavioural counselling, debriefing,. Control condition care as usual | 4–6 wks 12 wks | IES-R (PTSD) Traumatic birth defined as criterion A of PTSD DSM_V. | None reported | Results in favour of CBT counselling and debriefing over control condition at 4–6 wks and 12 wks. Sig diff between all groups at 4–6 wks and 12 wks (p < 0.001). No significant difference between the two intervention groups 4–6 weeks after the intervention. CBT counselling better than face to face debriefing at 12 weeks in reducing post-traumatic stress symptoms | 5 control, 6 CBT, 3 debriefing dropped out. Intervention content unclear.* DASS_21 (dep, anxiety, stress) Stress coping strategy scale utilized for inclusion. No secondary outcomes |
| Abdollahpour (2016) [42] | Hospital Iran | RCT N = 84 ITT 42:42 PP 39:39 | Women who had experienced an immediate traumatic birth before facilitation of the intervention. Qualified for PTSD criterion A DSM-5 as assessed by 4 item scale. | Midwife facilitated baby's natural instinctive response of 9 phases following birth. Intervention delivered in the hour following the traumatic birth event. | Care as usual | 4-6wks 12wks | IES-R | None reported | Sig diff in favour of intervention at 12 wks Reduction of PTSD symptomology. | No secondary outcomes |
| Asadzadeh (2020) [43] | Hospital Persia | RCT N = 90 PP 44:43 | Qualified for criterion A DSM-5 within 72hrs following birth. In the last trimester of pregnancy. Score over 10 Edinburgh PN Depression Scale | Midwife led counselling intervention based on Gamble's counselling intervention Dosage: 1 x face-to-face counselling session and 1 x telephone counselling session 4 to 6 weeks after giving birth. 1st author/ PhD researcher trained in and provided the intervention. Administered within 72 hrs following birth. | Care as usual | 4 wks 6 wks 12 wks | DSM-5-criterion A for the qualifying traumatic event "diagnosis of traumatic childbirth scale". Scale developed by the authors PCL-5 EPDS Hamilton anxiety rating scale | None reported | Results in favour of Midwife led counselling over care as usual at 4-6wks and at 12 weeks in reducing post-traumatic stress symptoms, depression, and anxiety. | |

*(Continued)*

**Table 3.** (Continued)

| Author | Country & Setting | Study Design No of Participants/ Interventions | Trauma Exposure | Intervention, Dosage & Timing | Comparison | Follow-up | Outcome Measures | Adverse Effects Participant Subjective Experience | Results | Comments |
|---|---|---|---|---|---|---|---|---|---|---|
| Chiorino (2019) [44] | Milan Italy Hospital setting | RCT N = 37 PP 19:18 | (Incl) Women have subjective experience of traumatic childbirth experience assessed subjectively and objectively by clinician. Score ≥24 on the (IES-R); | Brief one to one EMDR intervention utilizing the Birth Trauma Protocol. Dosage: 1 x 90 min session 90min Conducted within first 72 hrs following traumatic birth experience. Intervention delivered by clinical perinatal psychologist. | Care as Usual Standard supportive Psychological consultation focusing on emotions experienced during childbirth difficulties with caregiving, breastfeeding and psycho-physical recovery. Dosage– 1 x 90 min session Performed by 4x psychotherapists at MSc level or higher with supervision. | 6week and 12 week follow up by telephone. | PTSD symptoms IES-R Prevalence of participants asymptomatic IES-R score <23) PDEQ EPDS MIBS | None reported | Results in favour of intervention in reduced presence of flashbacks at 12 wks follow up (p = 0.042) Sig diff in overall PTSD symptoms in favour of intervention at 4- 6wks (p = 0.04) and 12 wk follow up (p = 0.03). Difference in number of women asymptomatic at 4–6 wks (p = 0.02) Cramers V = 0.408). No Diff at 12wks (p = 0.124) | MVAV suggests no interaction of outcome measures on effect size. Presence of flashbacks also measured. |
| Gamble (2005) [29] | Australia | RCT N = 103 PP 50:53 | Women reporting a traumatic birth experience as determined by Criterion A DSM-IV | Debriefing/face to face counselling. Dosage: 1 x face to face session of 40–60 mins within 72hrs following birth 1 x telephone session 4-6wks post-partum. Intervention delivered by midwife | Care as Usual | 4–6 weeks postpartum and 3 months postpartum | PTSD diagnosis and PTSD symptoms (MINI-PTSD) EPDS DASS-21, stress, anxiety | 86% women rated intervention highly (above 8/10) 90% Women reported that opportunity to talk about birth should be within a few days following birth. The remaining favoured 4 wks post-partum and, 2 x reported during pregnancy as the best time to discuss. | Results in favour of intervention at 3 months follow up. Sig diff in PTSD total symptom scores at 3-month follow-up Sig diff in Depression scores at 3 months follow up on the EPNDS X2 [1] = 9.188, p = 0.002 Sig diff in depression scores at 3 months follow up on the DASS-21 13(X2 [1] = 7.549, p = 0.005). Sif diff in stress scores at 3 months follow up on DASS-21 when compared with the control group (X2 [1] = 4.478, p = 0.029) | No statistical difference between groups in number of women meeting PTSD diagnosis at either 4 to 6 wks postpartum or three months postpartum. No significant difference in PTSD symptoms between groups at 4 to 6 weeks. |

(*Continued*)

**Table 3.** (Continued)

| Author | Country & Setting | Study Design No of Participants/ Interventions | Trauma Exposure | Intervention, Dosage & Timing | Comparison | Follow-up | Outcome Measures | Adverse Effects Participant Subjective Experience | Results | Comments |
|--------|-------------------|-----------------------------------------------|-----------------|-------------------------------|------------|-----------|------------------|---------------------------------------------------|---------|----------|
| Horsch (2017) [45] | Switzerland | RCT N = 56 ITT 29:27 PP 25:24 | Mothers recruited on the ward following EmCS | Taking part in a computerised visuospatial cognitive task within 6 hours following emergency caesarean section. Dosage– 10-15minute of play and daily diary of intrusive memories. Intervention delivered by midwife | Care as usual | 1 wk and 1 month | PTSD symptoms Posttraumatic Diagnostic Scale (PDS) ASDS stress HADS anxiety HADS depression Intrusive memories diary | Perceived to be acceptable by women. No reported harmful effects or serious incident was reported | No Sig diff between groups in post-traumatic stress disorder at one month (ITT) Sig diff favouring intervention in PDS avoidance cluster symptom count at 1 month (p = 0.05) Sig fewer intrusive traumatic memories at 1 week than control group (p = 0.017) Sig diff favouring intervention in reduced re-experiencing symptoms at 1wk ASDS (p = 0.06) No significant group differences in the ASDS total score or HADS Anxiety or Depression scores at 1wk or 1 month follow up. | Sig diff in PTSD diagnostic criteria at 1 month in per protocol analysis (p = 0.039) |
| Ryding (1998) [46] | Sweden | RCT N = 50: 49 completed | Women following EmCS recruited via a hospital obstetrics and gynaecology department No specific symptoms of PTSD | Counselling and psychoeducation intervention. Delivered between 8 days to 1-month post-partum. Delivered by Obstetrician with psychotherapy qualification. Dosage: 4 x sessions first consultation took ≥ 1 hour. The second to fourth meetings were 45 min. | Care as usual Discussion with midwife and Doctor who performed Em CS a few days after delivery. | 6 months post-partum | IES Prevalence of PTSD symptoms: mild <20 moderate 20–30 and severe (PTSD probable) score over 30 W-DEQ 20 fear of childbirth SCL 35 mental distress | None reported | Neutral No sig diff between groups in post traumatic stress symptoms at 6 months post partum. Sig Diff between groups in general distress at 6 months post partum. | No reported ethical approval. Median scores reported SD not reported. |

*(Continued)*

**Table 3.** (Continued)

| Author | Country & Setting | Study Design No of Participants/ Interventions | Trauma Exposure | Intervention, Dosage & Timing | Comparison | Follow-up | Outcome Measures | Adverse Effects Participant Subjective Experience | Results | Comments |
|---|---|---|---|---|---|---|---|---|---|---|
| Ryding [47] (2004) | Sweden | RCT N = 162:147 available at initial follow up | Women following EmCS recruited via a hospital obstetrics and gynaecology department Severity criterion: No specific symptoms of PTSD | Group counselling and education. Delivered 1–2 months post-partum Delivered by Psychologist and ward midwife. 4–5 women per group Dosage: 2 x 2hr sessions with 2–3 wk interval. | Care as Usual Invited to individual consultation with midwife following birth. | 6 months post-partum | IES Prevalence of post-traumatic stress symptoms IES>30 W-DEQ B EPNDS | Feedback from participants: Too few participants in the group. Needed additional sessions. Need to include fathers. | Neutral No sig diff between groups in childbirth experience, post traumatic stress or depression. | No reported ethical approval. Median reported No mean diffs or p reported. No baseline measures reported. |
| Shaw [48] (2013) | USA NICU | RCT N = 105 62:43 | Women who had developed symptoms of trauma, anxiety or depression following preterm birth. Recruited from NICU Women who scored above the clinical cut off on any instrument (BDI-II score over 20; BAI score over 16; SASRQ score over 3 for the required number of questions in 2 or more of the symptom categories) were invited into the intervention phase | TF-CBT and techniques to enhance parenting confidence twice a wk over 3-4wks. 2 x sessions conducted at bedside in NICU 4 x sessions in NICU Session 1: developing rapport Psychoeducation. Session 2: Cognitive restructuring Session 3 Progressive muscle relaxation Session 4: psychoeducation TF_CBT, session 5: trauma narrative session 6: infant redefinition. Intervention conducted by unlicensed Doctoral candidates completing PsyD. Dosage: 6 x 45-55min sessions | Active comparison of psychoeducation | 14 days following intervention, 4-5wks after birth, 6 months following birth. | DTS for DSM-IV SASRQ stress PSS:NICU BAI BDI Beck depression & anxiety Mini international neuropsychiatric interview (MINI-PTSD) | None reported | Sig moderate effect in favour of TF-CBT at 4–5 wk follow up in trauma symptoms [d = 0.41, p = 0.23] and depression [d = 0.59 p< .001] | Effect size between groups diffs before and after intervention not reported on. No power calculation Experimental and comparison groups unequal. SD not reported. |
| Slade (2020) [49] | UK Online | RCT N = 678 336: 342 | Women who reported their current birth experience as traumatic assessed by DSM-V criterion A | Psychological Self-help materials: A brief info leaflet and a web link to a film. Administered less than 3 months post-partum | Care as Usual | 6–12 weeks | Diagnostic and sub diagnostic PTSD CAPS-5 HADS anxiety & depression MPAS quality of attachment, hostility, pleasure in interaction DAS4 | No adverse effects Women viewed intervention favourably | Neutral | |

(*Continued*)

**Table 3.** (Continued)

| Author | Country & Setting | Study Design No of Participants/ Interventions | Trauma Exposure | Intervention, Dosage & Timing | Comparison | Follow-up | Outcome Measures | Adverse Effects Participant Subjective Experience | Results | Comments |
|---|---|---|---|---|---|---|---|---|---|---|
| ZelKowitz (2011) [50] | Montreal Canada Hospital and home Setting | RCT N = 121 48:50 | Mothers singleton infant born weighing less than 1500 grams recruited from NICU. PTSD symptoms related to experience of premature birth. No specifier. | 5x sessions in hospital (1–2 sessions per week over 3–5 wks) 1x telephone call 1 wk after discharge 1 x session at home 2–3 wks after discharge. Delivered by psychologist or nurse Total Dose 9–10 hrs Mother consents by 4th wk following birth | Usual Care and general information about caring for an infant | 4–6 weeks 6 months | PPQ Perinatal PTSD Questionnaire Not primary outcomes | None reported | Neutral | 48% of women intervention gp had PTSD scores in clinical range at baseline 52.5% women in CAU had PTSD scores within clinical range at baseline |

ITT, Intention to Treat; PP, Per protocol; EmCS, Emergency Caesarean section; Los Angeles Symptom Checklist (LASC) The Perinatal Risk Inventory (PERI), Depression, Anxiety and Stress Scale (DASS_21), Davidson Trauma Scale (DTS), The Stanford Acute Stress Reaction Questionnaire (SASRQ), Parental Stressor Scale: Neonatal Intensive Care Unit (PSS:NICU), The Beck Anxiety Inventory (BAI), Beck Depression Inventory (BDI), Mini-International Neuropsychiatric Interview–Post-Traumatic Stress Disorder (MINI-PTSD), Post Traumatic Stress Diagnostic Scale(PDS, Acute Stress Disorder Scale (ASDS), Hospital Anxiety and Depression Scale (HADS), Minnesota Multiphasic Personality Inventory-2 (MMPI-2), Perinatal PTSD Questionnaire (PPQ), Traumatic Event Scale (TES), Clinician Administered PTSD Scale for DSM-5 Diagnosis (CAPS-5). The Wijma Delivery Expectancy/Experience Questionnaire (W-DEQ form B), Impact of Event Scale & Impact of Event Scale Revised (IES & IES-R), Edinburgh Postnatal Depression Scale (EPDS), Peritraumatic Dissociative Experiences Questionnaire (PDEQ), Mother to Infant Bonding Scale (MIBS), Symptoms Check List (SCL) Multidimensional Parental Attachment Scale (MPAS), Dyadic Adjustment Scale, (DAS4).

**3.3.1 Random sequence generation (selection bias).** Eight studies provided an adequate description of the randomisation process and were at low risk of selection bias [29, 41–45,48, 49]. One study was unclear risk of allocation concealment [50] Two studies were at high risk of selection bias as the process was not entirely random [46, 47]. Women who gave birth on "approximately 18 predetermined days of the month" were randomised to the counselling group [46]. One study selected every second woman who had experienced an emergency cae-sarean, according to the delivery ward register, for the counselling intervention, with the remainder selected for the comparison group [47].

**3.3.2 Allocation concealment (selection bias).** Six studies reported adequate conceal-ment procedures and were at low risk of selection bias [29, 41, 42, 44, 45, 49]. In three studies, allocation concealment was unclear [43, 48, 50]. Two studies did not hide allocation conceal-ment and were at high risk of selection bias [46, 47].

**3.3.3 Blinding of outcome assessment (detection bias).** Blinding of participants and per-sonnel was not assessed in this review as a double-blind methodology for studies of psycholog-ical treatment is not possible. It is clear to participants what treatment they are receiving. Seven studies blinded outcome assessors and were at low risk of detection bias [43, 44, 46, 49, 47, 48, 50]. Two studies it was not clear whether assessors of all reported PTSD outcomes were blinded to group allocation [29, 45] and two studies were at high risk of detection bias as blind-ing of outcomes assessors was not reported on at all [41, 42].

**3.3.4 Incomplete outcome data (attrition bias).** Five studies fully reported loss to follow-up and adequately dealt with missing outcome data [29, 45, 47–49]. Horsch [45] and Slade [49] calculated Intention to treat [ITT] analysis. Shaw [48] reported that they treated

| Intervention type | Intervention description | Facilitated by |
|---|---|---|
| Immediate response early interventions | 1. Self-help psychoeducation materials in print and online video format<br>2. Facilitation of the baby's natural instinctive response of 9 phases in the first hour following childbirth<br>3. Computerised visual spatial cognitive task administered by a midwife<br>4. Cognitive behavioural counselling<br>5. Debriefing counselling<br>6. Trauma informed counselling<br>7. Single session EMDR early intervention | Self-led<br>Midwife<br><br>Midwife<br>Midwife<br>Midwife<br>Midwife<br>Clinical Psychologist |
| Group based early interventions | Group counselling | Midwife & Psychologist |
| High intensity early interventions | 1. Trauma focused cognitive behavioural therapy (TF-CBT)<br>2. Care and Cues behavioural education intervention<br>3. Counselling and psychoeducation | Doctoral students<br>Psychologist<br>Obstetric doctor |

**Fig 2. Early psychological interventions targeting PTSD in women following traumatic childbirth.**

unavailable data as missing at random "conditional on observed information" according to a maximum likelihood estimation. Gamble [29] included follow-up data from all participants at 12-week outcomes. No studies calculated an estimation of outcome by the method of 'last observation carried forward'. Six studies that provided data for nearly all participants randomised and reported reason and number of withdrawals were at unclear risk of bias [41–44, 46, 50]. No studies were at a high risk of attrition bias.

**3.3.5 Selective reporting (reporting bias).** Two studies reported analysed data to a pre specified protocol and were at low risk of reporting bias [45, 50]. It was unclear if the remaining studies were free from reporting bias as they did not publish pre-specified study protocols [41–44, 46, 47, 48, 49]. It was unclear as to whether four studies investigated PTSD symptomology or diagnosis as a primary outcome [29, 43, 46, 47] and were therefore at unclear risk of reporting bias.

**3.3.6 Other bias.** Three studies were at low risk of other bias [43, 49, 50]. Five studies were at unclear risk of other bias [41, 42, 45–47] as treatment fidelity was not reported. Three studies were at high risk of other bias as the study authors developed the intervention protocol [29, 44, 48].

## 3.4 Effects of intervention

Continuous outcomes were analysed using mean difference (MD) in scales assessing for PTSD. Three studies used the same scale to measure symptoms of PTSD [37–39]. The standardised mean difference (SMD) was used as the summary statistic in meta-analysis [51] as other studies measured PTSD using differing psychometric scales. Dichotomous data were managed with the risk ratio (RR) as the main categorical outcome measure. All outcomes were presented using 95% confidence Intervals (CI). Rules for interpreting SMDs (or 'Cohen's effect sizes') were guided by Cohen (1988) [52] as <0.40 = small, 0.40 to 0.70 = moderate, >0.70 = large. (Please refer to Table 4 for summary of meta-analysis of results and GRADE ratings).

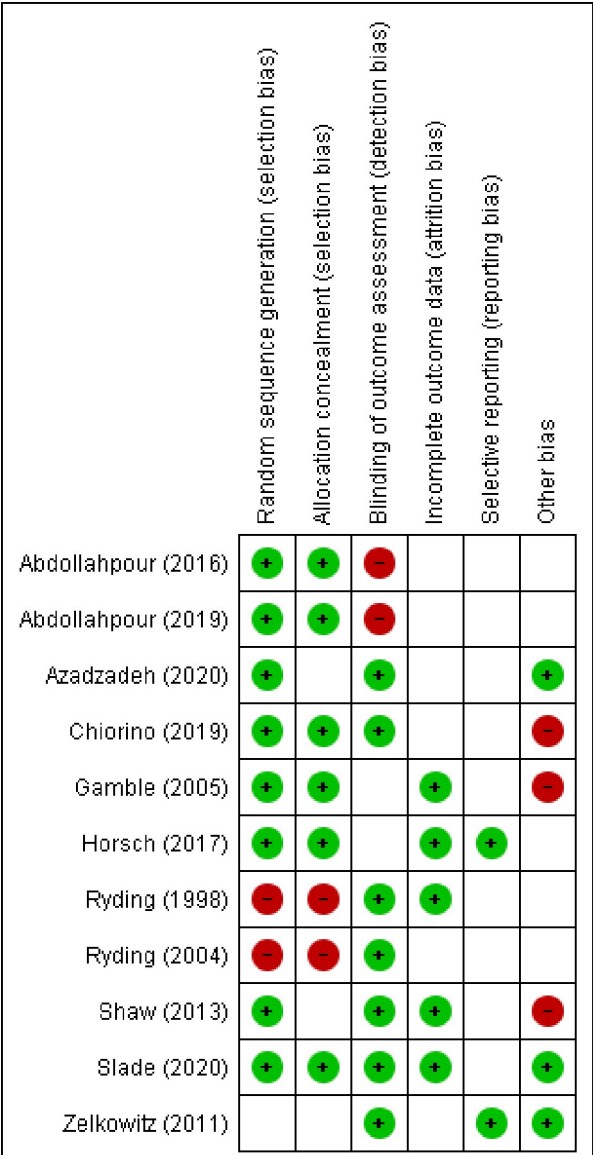

**Fig 3. Risk of Bias summary table.**

Three studies continued the assessment process with participants who dropped out of a randomly assigned treatment control or intervention group [45, 48, 49]. These studies were able to report on true intention to treat. All other studies reported per protocol results of mean differences and did not continue assessments with participants who were excluded from analysis or lost to follow up [29, 41–44, 46, 47, 50].

**3.4.1 Outcome: PTSD symptom severity 4–6 weeks post-partum.** Six studies included data for assessing the severity of PTSD symptoms 4–6 weeks post-partum [29, 41–45]. Debriefing intervention from the three-armed trial [41] was excluded from the meta-analysis to avoid a unit of analysis error. The "combining groups" approach was not applied to the pairwise random effects model to avoid clinical methodological error between the debriefing and cognitive behavioural counselling modes of intervention (Chi$^2$ = 17.78, df = 1 (P < 0.0001); I$^2$ = 94%). Meta- analysis showed evidence of a difference between early psychological interventions and

**Table 4. Summary of meta-analysis.** Table of results for early interventions administered within one month of exposure to a traumatic birth.

| Comparison | Follow-up and contributing studies | Study (no) | Sample (n) | Relative Risk (95% CI) | Std.Mean Diff (95% CI) | Grade Rating |
|---|---|---|---|---|---|---|
| **Interventions delivered within 72hrs following traumatic birth** Immediate response early psychological Intervention vs CAU Dosage 1–2 sessions | PTSD symptom severity 4–6 weeks post-partum Abdollahpour, 2016; Abdollahpour 2019; Azadzadeh 2020; Chiorino 2019; Gamble 2005; Horsch, 2017 [29, 41–45] | 6 | 224:258 | | PP -0.58 | Moderate |
| | PTSD symptom severity 12wks weeks post-partum Abdollahpour, 2016; Abdollahpour 2019; Azadzadeh 2020; Chiorino 2019; Gamble 2005 [29,41–43] | 5 | 199:234 | | PP -1.08 | Low |
| Midwifery led debriefing vs CAU Dosage 1 x session | PTSD symptom severity 12 weeks post-partum Abdollahpour, 2019 [41] | 1 | 51:81 | | PP -0.84 | Low |
| TF-CBT counselling vs active intervention debriefing Dosage 1 x session | PTSD symptom severity 12 weeks post-partum Abdollahpour, 2019 [41] | 1 | 47:51 | | PP -1.45 | Low |
| Early EMDR vs CAU Dosage 1xsession | PTSD Rates remission 6 weeks post-partum Chiorino 2019 [44]. PTSD Rates remission 12 weeks post-partum Chiorino 2019 [44]. | 1 | 19:18 | 2.03 p = 0.03 1.34 P = 0.11 | | Low |
| Visuospatial gaming activity Dosage 1xgame Tetris | Prevalence of intrusive memories 1-week post-partum Horsch 2017 [45]. | 1 | ITT 29:27 PP 25:24 | | ITT 0.41 PP -5.46 | Low |
| Early EMDR vs CAU Dosage 1x session | Prevalence of intrusive memories 12 weeks post-partum Chiorino 2019 [44]. | 1 | 19:18 | 0.16 P = 0.07 | | Low |
| Midwifery led brief counselling intervention Dosage 2 x sessions | Diagnosis of PTSD 4–6 weeks post-partum Gamble, 2005 [29]. Diagnosis of PTSD 12 weeks post-partum Gamble, 2005 [29]. | 1 | 50:53 | 1.13 P = 0.68 0.33 P = 0.08 | | Low |
| **Interventions delivered up to 3 wks following traumatic premature birth** Early TF-CBT therapy vs active psychoeducation intervention Dosage 6–9 sessions | PTSD symptom severity 4–6 weeks post-partum Shaw, 2013 [48]. | 1 | 62:43 | | ITT -0.10 d = -0.33 adjusted Bonferroni d = 0.41 p = .023 | Low |
| **Intervention delivered 33 days following traumatic premature birth** Cues intervention vs Attention Control Comparison Dosage 6 x sessions Cues Programme Dosage 6 x contacts with care intervener (active comparison) | PTSD symptom severity: follow up time unclear Zelkowitz,2011 [50]. | 1 | 48:58 | | -0.10 | Low |
| **Intervention self-administered within 4 weeks of a traumatic birth** Online recorded footage and paper-based Psychoeducation Dosage unlimited self-led | Clinical criteria for diagnosis of PTSD 12 weeks Slade, 2020 [49]. | 1 | 336:342 | ITT 0.77 p = .40 PP 0.81 p = .67 | | Moderate |

treatment as usual (SMD -0.58, 95% CI -0.91, -0.26; $I^2$ = 66% studies = 6 No = 224:258). There was a moderate degree of heterogeneity between studies as can be seen in the forest plot (Fig 4). Medium GRADE quality evidence was reflected by high risk of methodological bias. Sensitivity analysis was performed for cultural differences. Removal of data from three studies conducted in Iran reduced heterogeneity to $I^2$ = 0% Sensitivity analysis between the two studies

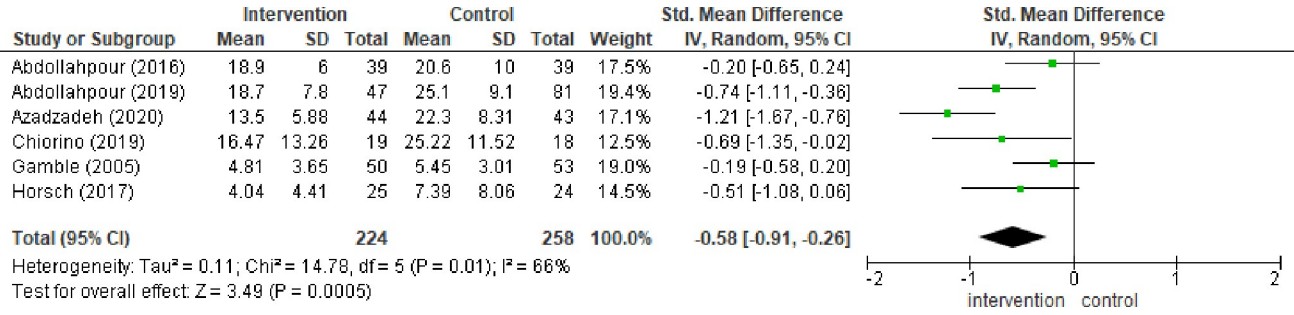

**Fig 4. Forest plot: Symptom severity 4–6 weeks post-partum.**

assessing the effectiveness of Gambles counselling technique [29, 43] resulted in a high degree of heterogeneity between the two studies ($I^2$ = 91%). No baseline differences were reported between groups [41].

Forest plot depicting difference between psychological intervention and treatment as usual in PTSD symptom severity in women 4–6 weeks post-partum (SMD -0.58, 95% CI -0.91, -0.26; $I^2$ = 66% studies = 6 No = 224:258).

**3.4.2 PTSD symptom severity 12 weeks post-partum.** Five studies provided analysis for PTSD symptom severity at 12 week follow up [29, 41–44]. Random effects meta-analysis (Fig 5) found a strong effect in favour of early psychological interventions when compared with usual care (SMD -1.08 95% CI 1.67, -0.49). Clinical, cultural and methodological heterogeneity was reflected by the statistic ($I^2$ = 87%). Variance in type of intervention and measurement was reflected in wide confidence intervals and imprecision. Grade rating was marked up by 1 level for magnitude of effect. Grade quality evidence was low.

Forest plot depicting difference between psychological intervention and treatment as usual in PTSD symptom severity in women 12 weeks post-partum (SMD -1.08 95% CI 1.67, -0.49; $I^2$ = 87% studies = 5 No = 199:234).

Abdollahpour [41] conducted a three-armed study that compared midwifery led cognitive behavioural counselling and debriefing with routine post-partum care without counselling or psychological intervention. Wide confidence intervals for debriefing (SMD -0.84 95% CI -1.21, -0.48) at 12 weeks follow up resulted in Grade quality evidence low. Magnitude of effect was reflected in grade quality evidence marked up by 1 level. Single session cognitive behavioural counselling was more effective than debriefing at 12 weeks post-partum (SMD -1.45 95% CI -1.89, -1.00).

A three-armed parallel design study assessed a high intensity trauma focused CBT intervention with a total of six sessions. Shaw [48] measured Post traumatic stress symptoms in women

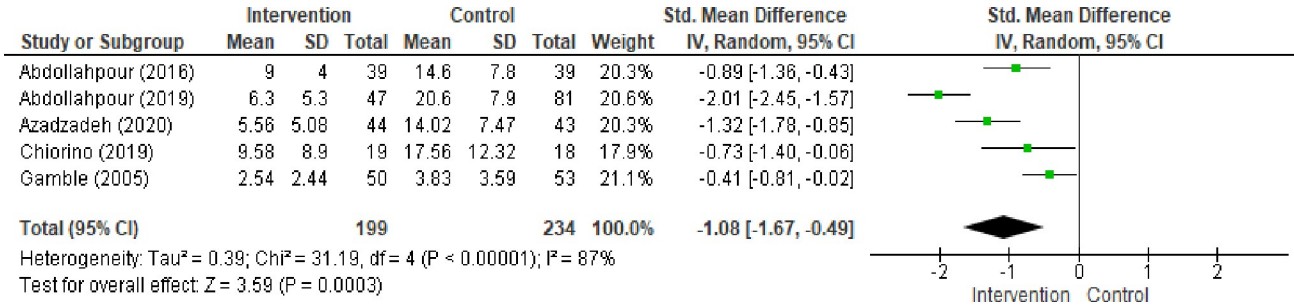

**Fig 5. Forest plot: Symptom severity 12 weeks post-partum.**

who experienced a traumatic preterm birth. Women were randomised to trauma focused CBT intervention and active psychoeducational comparison at baseline and one week following completion of the intervention. The study did not report SD so was not included in the meta-analysis. Authors reported ITT analysis and standard linear mixed effects modelling, estimating that a psychologist led trauma focused CBT program with parenting education was more effective than psychoeducation in reducing post-traumatic stress symptoms (M = 35 [CI 95% 30–40] vs M = 37 [CI 95% 30–45] p = .02) at 4 to 5 weeks post-partum. There was a moderate effect of the intervention on scores of post-traumatic stress symptoms at 4 to 5 weeks following birth (Cohen's d = 0.41, P = .023). Grade quality evidence was low due to risk of bias and unequal participant numbers in intervention and control conditions may have inflated results and type II error. Another study did not find any difference between standard care and a high intensity psychoeducational and cognitive behavioural parenting intervention in reducing PTSD symptoms in women who had experienced the trauma of premature birth and their baby's admittance to the NICU [50].

A counselling intervention [47] was more effective than control condition in reducing PTSD symptomology in women who had experienced an emergency caesarean, one month following the intervention (Median = 3.0 vs Median = 7.5 p = .01) but not at six months post-partum. Standard deviations or confidence intervals were not reported so this study was not included in the overall meta-analysis for PTSD symptom severity. Ryding [46] found no difference between women in a group counselling intervention and care as usual in symptoms of post-traumatic stress (median Impact of Event Scale score 12.0 vs 10.0), at 6 months post-partum. Mean scores or confidence intervals were not reported, so this study was not included in the meta-analysis for PTSD symptom severity.

**3.4.3 PTSD diagnosis.** Dichotomous data reported on rates of PTSD remission at 4 to 6 weeks follow up, by one RCT [47] and one pilot study [44], could not be analysed in a meta-analysis as there was clinical heterogeneity between the cut off scores that constituted as normal or mild; and moderate PTSD symptomatology. One study [44] reported the prevalence rates of women who scored less than 20 as in remission at 6 weeks following traumatic birth (RR 2.03 95%CI 1.09, 3.79) with low certainty in the evidence and low certainty of evidence at 12 weeks on asymptomatic profiles between the group that received EMDR and the control group (RR 1.34 95% CI 0.94, 1.93) as reflected by wide confidence intervals, small sample sizes and risk of bias.

A study investigating the effects of self-help materials [49] found no difference in rates of full diagnosis between women who had received self-help materials and those who had received usual care 6 to 12 weeks post-partum. Intention to treat analysis was calculated (RR 0.77 95% CI 0.41,1.43 Z = 0.84 [p = .40]) as was per protocol analysis (RR 0.81 95% CI 0.31, 2.12 Z = 0.43 [p = .67]). The authors' ITT analysis were consistent with per protocol analysis, which excluded women who were randomised in error by research midwives (RR 1.04, 95% CI 0.85–1.27) with ITT (RR 1.02, 95% CI 0.68–1.53, P = 0.92) with moderate GRADE certainty in the evidence.

**3.4.4 Depression symptom severity.** Seven studies reported on women's symptoms of depression following a traumatic birth experience [29, 43–45, 47–49]. These can be viewed on the study characteristics Table 3. There was insufficient data available to perform meta-analysis on symptom severity of depression. We will report on individual study level analysis of difference and significance. There was no significant difference between women who received one session of EMDR versus standard care as usual [44] in symptoms of post-natal depression at 6 and 12 week follow up. A significant difference in favour of midwifery counselling versus usual care was found at 4–6 weeks and 12 weeks post treatment [29, 43] along with psychologist delivered TF-CBT versus active control at 4–5 week follow up [48]. There was no

significant difference between women who received 2x sessions of group counselling compared with usual care in symptoms of depression at 6 months follow up [47]. No significant difference was found between performing a computer game of Tetris in taxing working memory and usual care control in symptoms of depression in women who had experienced a caesarean section at one week and one month follow up [45].

## 4.0 Discussion and implications for practice

This is the first review, to our knowledge that assesses the effectiveness of early psychological interventions for the prevention and treatment of PTSD and respective symptoms in women during the immediate period following a psychological or physically traumatic birth. The methodological quality of the studies was variable. Studies were culturally diverse, and interventions varied in their theoretical basis and mechanisms of change in pathogenesis of PTSD.

Studies that provided sufficient data for meta-analysis of PTSD symptom severity at 4–6 week and 12 week follow up included low intensity early counselling interventions [29, 41, 43] and low intensity early psychosensory interventions at 4–6 weeks [42, 44, 45]. Only two of these intervention types provided data for 12 weeks follow up [42, 44]. All studies assessed secondary prevention of PTSD and PTSD symptomology.

There is evidence on a per protocol basis that immediate response early midwifery or clinician led interventions (1–2 sessions) delivered within 72 hours of a traumatic birth are more effective in reducing symptoms of PTSD in women who have experienced a traumatic birth than care as usual at 4–6 weeks post-partum (SMD -0.58, 95% CI -0.91, -0.26; $I^2$ = 66% studies = 6 No = 224:258). Reduced symptoms are sustained up to 12 weeks postpartum (SMD -1.08 95% CI 1.67, -0.49; $I^2$ = 88% studies = 5, No = 199:234). These results shed light on the outcomes of women who have been screened for the birth as a qualifying criterion A psychologically traumatic event, in the early period following exposure to a traumatic birth.

These results are in parallel with a recent review [32] that highlighted the benefit of both trauma focused CBT and EMDR administered to adults reporting clinically significant traumatic stress symptoms. These results also add to the body of evidence as documented by the ISTSS [21] who report emerging evidence for single session EMDR within the first three months of a traumatic event for the prevention and treatment of PTSD symptoms in adults. Recent research has found that skin to skin contact is a protective factor, guarding against post-traumatic stress symptoms in women 1 to 5 years following normal childbirth [53]. This mode of intervention in the immediate period following a traumatic birth maybe clinically difficult to facilitate given the dyadic effects of a traumatic birth or premature birth on both mother and infant.

There were mixed results from the three studies assessing the effectiveness of high intensity early psychological interventions following an emergency caesarean [47] and traumatic premature birth experience [48, 50]. Insufficient data was available for performing meta-analysis of high intensity early psychological intervention. Further studies are required to investigate the effectiveness of early interventions in reducing sub clinical or clinically significant symptoms of post-traumatic stress in this population. There were also mixed results from the seven studies reporting on the effectiveness of early psychological intervention on secondary outcome of postpartum depression in women following a traumatic birth experience. Further studies are required to provide more conclusive evidence on the effects of early psychological intervention on symptoms of depression given the high co morbidity and multimorbidity between PTSD, depression and physical effects of traumatic birth.

## 4.1 Limitations and recommendations

This literature review includes investigation of early psychological interventions in the form of exposure and non-exposure-based trauma informed interventions and novel interventions that are built upon psychological theoretical concepts and neuroscientific findings in response to a traumatic birth [54, 55]. Randomised controlled trials and randomised pilot studies investigating early interventions informed by psychological theory were included. Results indicated that a self-directed online and paper based psychoeducational intervention is no more effective than standard care in preventing post-traumatic stress disorder. This suggests that other modes of midwifery or clinician led interventions are required to prevent the disorder. RCTs that focused on treating or preventing post-traumatic stress symptoms due to the experiences of the Neo natal intensive care unit alone were not included in this analysis. Davis and Stein [56] documented women's personal pre-term birth experiences. Analysis of women's verbatim suggests that mothers' symptoms of PTSD and associated mental illness may not be singly attributed to the experience of the NICU environment. The scale of traumatic events that take place in the short space of time between birth or premature birth, admittance to and discharge from the NICU are enveloped within a full traumatic episode. This series of events make up the whole. It is recommended that further interventions aimed at targeting PTSD symptoms in women whose infants are admitted to the NICU include assessment of symptoms of PTSD attributed to the prenatal and actual birthing experience.

High variability and heterogeneity between the immediate response early psychological interventions at 12 weeks post-partum indicate that caution should be taken when interpreting the effectiveness of early intervention meta-analysis results. The high degree of heterogeneity between the two studies assessing Gambles counselling intervention [29, 43] may be due to cultural differences (Iran/Australia) and or variation between the measures used to assess post-traumatic stress symptoms (PCL-5 and Mini-PTSD). This also demonstrates assessment of intervention to fidelity as an important aspect of RCT design, enabling evidence on the reliability and validity in intervention delivery.

A group counselling intervention [47] administered at one to two months post-partum was no more effective than standard care in reducing symptoms of PTSD in women who had experienced a caesarean section at six months follow up. Qualitative feedback from women suggest that more than two sessions would have been necessary as discussing the birth left the experience "open". Further studies assessing group interventions may administer the intervention in the immediate time period following the birth, include additional sessions, or apply a blind to therapist protocol whereby women are not required to openly discuss their experiences, thoughts or feelings [57].

Early interventions that work to integrate implicit and unconscious traumatic birth memories and neural pathways within the brain, may be particularly helpful to women whose trauma maybe shame based [58–60]; interwoven with previous traumatic material and adverse childhood experiences. Immediate response early intervention may help stabilise and reduce a wide range of symptoms, with an extension of more extensive therapy that may be necessary to prevent full PTSD diagnosis.

Results suggest that both top down and bottom up processing of both episodic and implicit traumatic memories may be helpful for women 72 hours following a traumatic birth. Further studies could assess for PTSD, complex PTSD and dissociative subtype, as this factor could also account for heterogeneity [61–63]. In light of this, it is recommended that transdiagnostic modes of early intervention are explored as prevention and treatment modalities for traumatic stress symptoms and PTSD in women following a traumatic birth experience.

Interventions included in meta-analysis are similar in that all are administered within 72hours following childbirth. This factor may account for the effectiveness of these early

psychological interventions, as it may be theoretically important for unconsolidated memories to be efficiently integrated within the memory network in the days immediately following trauma exposure in real time.

There were no studies assessing early psychological interventions, set within the parameters of the review search criteria, that utilised biomarkers to assess the effects of a traumatic birth. Women's risk of developing PTSD following a traumatic birth due to neuro endocrine factors [64] indicate that an intervention with psychobiological evidence may be particularly appropriate for this population.

## 5.0 Conclusion

There is firm evidence in favour of immediate response early psychological interventions administered within 72 hours of a traumatic birth in reducing symptoms of PTSD in women on a case by case basis. Further long-term studies are required in order to address methodological weaknesses before recommendation can be made to clinical practice. Future studies that target clinical diagnosis of PTSD and co morbidities are necessary. Multifinality of risk factors associated with the development of PTSD in post-partum women, such as experiencing the birth as traumatic and having previous experience of traumatic birth in parous women are important when determining appropriate application of early psychological intervention for women in the perinatal period; applying clinical equipoise to women's needs and reducing unnecessary costs.

## Supporting information

**S1 File. PRISMA checklist.**
(PDF)

## Acknowledgments

We wish to express our thanks to the Life and Health Sciences subject librarian Kelly Mc Koo at Ulster University, Northern Ireland for providing reliability and validity to the database searches. We also wish to thank Jennifer Allen and William Mc Murray for their assistance in locating documents.

## Author Contributions

**Conceptualization:** P. G. Taylor Miller, M. Sinclair, P. Gillen, J. E. M. McCullough, P. W. Miller, D. P. Farrell, P. Klaus.

**Data curation:** P. G. Taylor Miller.

**Formal analysis:** P. G. Taylor Miller.

**Investigation:** P. G. Taylor Miller.

**Methodology:** P. G. Taylor Miller.

**Project administration:** P. G. Taylor Miller.

**Supervision:** M. Sinclair, P. Gillen, P. W. Miller, D. P. Farrell.

**Validation:** M. Sinclair, P. Gillen, J. E. M. McCullough, P. W. Miller, P. F. Slater, E. Shapiro, P. Klaus.

**Visualization:** P. G. Taylor Miller.

**Writing – original draft:** P. G. Taylor Miller.

**Writing – review & editing:** P. G. Taylor Miller, M. Sinclair, P. Gillen, J. E. M. McCullough, P. W. Miller, D. P. Farrell, P. Klaus.

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
