## [Decision Letter · Decision Letter 0]

21 Sep 2021

Early psychological interventions for prevention of post-traumatic stress disorder (PTSD) and treatment of post-traumatic stress symptoms in post-partum women: A systematic review and meta-analysis

PONE-D-21-20426

Dear Dr. Miller,

We’re pleased to inform you that your manuscript has been judged scientifically suitable for publication and will be formally accepted for publication once it meets all outstanding technical requirements.

Kind regards,

Vedat Sar, M.D.

Academic Editor

PLOS ONE

Additional Editor Comments (optional):

Reviewers' comments:

Reviewer's Responses to Questions

**Comments to the Author**

1. Is the manuscript technically sound, and do the data support the conclusions?

Reviewer #1: Yes

2. Has the statistical analysis been performed appropriately and rigorously? 

Reviewer #1: Yes

3. Have the authors made all data underlying the findings in their manuscript fully available?

Reviewer #1: Yes

4. Is the manuscript presented in an intelligible fashion and written in standard English?

Reviewer #1: Yes

5. Review Comments to the Author

Reviewer #1: 1. The manuscript is technically sound and supports the conclusion

2. The statistical analysis has been performed appropriately

3. Sufficient details of methods and analysis has been provided

4. The conclusions are drawn adequately

5. Figures and tables are clearly presented

6. PLOS authors have the option to publish the peer review history of their article (what does this mean?). If published, this will include your full peer review and any attached files.

Reviewer #1: **Yes: **Suman Prasad Adhikari

---

## [Editor Report · Acceptance letter]

9 Nov 2021

PONE-D-21-20426 

Early psychological interventions for prevention and treatment of post-traumatic stress disorder (PTSD) and post-traumatic stress symptoms in post-partum women: A systematic review and meta-analysis 

Dear Dr. Taylor Miller:

I'm pleased to inform you that your manuscript has been deemed suitable for publication in PLOS ONE. Congratulations! Your manuscript is now with our production department. 

Kind regards, 

on behalf of

Dr. Vedat Sar 

Academic Editor

PLOS ONE